# communications
## engineering

# Operando two-terminal devices inside a transmission electron microscope

Oscar Recalde-Benitez [1], Tianshu Jiang[1], Robert Winkler[1], Yating Ruan[2], Alexander Zintler [1], Esmaeil Adabifiroozjaei [1], Alexey Arzumanov [2], William A. Hubbard[3], Tijn van Omme [4], Yevheniy Pivak[4], Hector H. Perez-Garza [4], B. C. Regan [3,5], Lambert Alff [2], Philipp Komissinskiy [2] & Leopoldo Molina-Luna [1✉]

Advanced nanomaterials are at the core of innovation for the microelectronics industry. Designing, characterizing, and testing two-terminal devices, such as metal-insulator-metal structures, is key to improving material stack design and integration. Electrical biasing within in situ transmission electron microscopy using MEMS-based platforms is a promising technique for nano-characterization under *operando* conditions. However, conventional focused ion beam sample preparation can introduce parasitic current paths, limiting device performance and leading to overestimated electrical responses. Here we demonstrate connectivity of TEM lamella devices obtained from a novel electrical contacting method based solely on van der Waals forces. This method reduces parasitic leakage currents by at least five orders of magnitude relative to reported preparation approaches. Our methodology enables operation of stack devices inside a microscope with device currents as low as 10 pA. We apply this approach to observe in situ biasing-induced defect formation, providing valuable insights into the behavior of an $SrTiO_3$-based memristor.

[1] Advanced Electron Microscopy Division, Institute of Materials Science, Department of Materials- and Geosciences, Technische Universität Darmstadt, Darmstadt, Germany. [2] Advanced Thin Film Technology Division, Institute of Materials Science, Department of Materials- and Geosciences, Technische Universität Darmstadt, Darmstadt, Germany. [3] NanoElectronic Imaging, Inc., Los Angeles, CA, USA. [4] DENSsolutions, Delft, The Netherlands. [5] University of California, Los Angeles and the California NanoSystem Institute, Los Angeles, CA, USA. ✉email: leopoldo.molina-luna@aem.tu-darmstadt.de

Despite the high spatial resolution achievable with modern aberration-corrected scanning transmission electron microscopes (STEM)[1], atomic-scale and electronic structure characterization of functional devices under an applied electric field (E) under operative conditions has been a challenging task. The capability to perform electric current-voltage (I–V) measurements of cross-section electronic devices inside an electron microscope while simultaneously analyzing the corresponding structural, chemical, or even electronic structure changes in so-called *operando* conditions would be a breakthrough in the field of nanoelectronics. The idea of applying an external biasing stimulus using electrical contacts for in situ TEM was introduced in 1967[2]. However, recent technological advances involving microelectromechanical systems (MEMS)-based chips or moving probes have opened promising new avenues.

The moving probes method has demonstrated outstanding electrical response for two-terminal devices, but it is primarily used to locally inject current, creating an inhomogeneous electric field that can change the local functional properties of materials and interfaces in thin-film heterostructures for nanoelectronics[3–5]. Furthermore, the use of probe tips limits the in situ TEM capabilities to the application of only one stimulus at a time, precluding the study of multi-stimulus dynamic processes. On the other hand, MEMS-based chips usually consist of two or more electrodes patterned on a highly resistive silicon nitride substrate, enabling measurement configurations with various stimuli, such as heating, electrical bias, gas flow, etc., applied either separately or in combination. Moreover, operando TEM studies with homogenous applied stimuli distributed across a TEM *lamella* is now feasible using MEMS-based chips.

Up to now, operando TEM studies of electrically biased *lamellae* using MEMS chips were typically employed at relatively low magnification[6], with sub-nanometer resolution achieved in only a few cases[7–11]; revealing electric field-related phenomena but not necessarily implying device *operando* conditions. Independent of spatial resolution, I–V measurements on scaled-down/micron-scale samples with thicknesses of less than 100 nm (maximum/required thickness for an electron transparent TEM sample) are technically challenging due to complex multi-step sample preparation that includes micromanipulation and electrical contacting. The use of conductive materials such as carbon, platinum, or tungsten with electron (EBID) and/or ion beam induced deposition (IBID) to attach and/or contact the sample via a FIB gas injection system (GIS) on MEMS-based chips has been, to date, the standard practice in the electron microscopy community[7–14]. However, FIB (GIS)-deposited materials can spread over tens of microns on a MEMS chip surface[13], forming undesirable stray leakage current pathways between the chip leads and the surroundings, affecting the TEM *lamella* itself (Fig. S1)[15]. For instance, for the 200 nm-thick GIS-deposited Pt and C pathways in the empty MEMS-based chip (no *lamella*) shown in Fig. S2, parasitic currents in the range of $10^{-5}$ A were measured between the inner and outer electrodes at an applied dc bias of 0.1 V. Similarly, the leakage current values reported for two-terminal thin-film metal-insulator-metal (MIM) TEM *lamella* devices range from $10^{-5}$ to $10^{-2}$ A at 0.1 V[7–10,14,16–18] (Figs. S3 and S4). Interestingly, to our knowledge no *operando* TEM studies using MEMS-based chips of MIM *lamella* devices with currents lower than $5·10^{-6}$ A at 0.1 V have been reported in the literature. Considering that in these cases, electrical testing may actually be measuring shunting in the *lamella* devices due to GIS-deposited conductive layers rather than revealing the actual electronic performance of the MIM device, a comparison of the short-circuit state between different material systems is valid. The shunting of TEM *lamella* devices during sample preparation has enormous implications, affecting not only the electric responses

of MIM devices, but also the in situ TEM observations, as observed microscopic behavior might not be intrinsic to the studied material, but rather an effect of short-circuiting on the MEMS chip (Fig. S5).

## Results and discussion

**FIB-based electrical contacting of two-terminal devices**. In this study, we present evidence of the operability of MIM devices utilizing an innovative FIB sample preparation technique employing MEMS-based chips[12,19,20]. Our FIB methodology eliminates the occurrence of undesired short-circuits, allowing reliable electric contacting and realistic I–V measurements of *lamellae* of electrode-based thin-film oxide electronic devices inside a TEM. In this approach, a TEM *lamella* is attached directly onto the electrodes of a MEMS chip and then contacted without employing GIS-deposited conductive layers, as shown in the step-by-step routine described in Fig. 1. Note that the sample preparation routine is similar to those used for conventional FIB-based TEM *lamellae* fabrication, starting with the deposition of a protection layer (Fig. 1A)[12,21]. However, some fundamental differences must be emphasized. For example, during the undercut process two lateral cuts are made to isolate the bottom and top electrodes of the device stack[7,14] (Fig. 1B), followed by a standard lift-out process (Fig. 1C). Additionally, the TEM *lamella* is thinned to its final thickness (<100 nm) directly on the FIB micro-manipulator to avoid any damage of the MEMS chip (Fig. 1D). Finally, the *lamella* is then attached to the chip leads by van der Waals forces solely. The side and front-view images of the sample in Fig. 1E, F shows proper contact with the chip leads. Our approach enables measurement of electronic responses in TEM *lamellae* of two-terminal MIM nano-devices with a resolution below 10 pA (Fig. 1G). The obtained low current values fit with the currents expected for a MIM device scaled to TEM *lamella* dimensions (Fig. S7), allowing for reliable *operando* TEM studies. Furthermore, our FIB-based approach has a high reproducibility rate, as shown in Fig. 1H where four TEM *lamella* devices extracted from the same millimeter-sized device (STO-based memristor) yield similar I–V trends. The time required to prepare a sample was significantly reduced, approximately half of the time, thanks to the elimination of steps related to sample attachment and the establishment of a conductive path using the GIS system. Therefore, we believe this process could open the doors for routine in situ biasing and further applications in multi-stimuli experiments.

**Electrical response of macro vs. TEM *lamella* devices**. For this study four types of two-terminal MIM nanodevices with epitaxially grown thin-film oxides were fabricated: (a1) $Pt/SrTiO_3/Nb$-$SrTiO_3$ and (a2) $Pt/Sm$-$CeO_2$:$SrTiO_3/Nb$-$SrTiO_3$ memristors[22], (b1) $Au/Pt/Ba_{0.5}Sr_{0.5}TiO_3/SrMoO_3$[23–25] and (b2) $Au/Pt/Mn$-$Ba_{0.5}Sr_{0.5}TiO_3/SrMoO_3$ tunable capacitors (varactors) (Table S1). TEM *lamella* devices were prepared by FIB and their electrical properties were measured and compared to the corresponding macroscopic counterparts (Fig. 2A and Figs. S8–S10).

Measurements of electrical conductance between the electrodes of an empty MEMS chip revealed a leakage current below 10 pA through the silicon nitride substrate, which corresponds to the lowest reliably measurable current of a TEM *lamella* device in our setup (dashed yellow line in Fig. 1G) (see also Fig. S12). The electrical measurements of memristor (a1) at low applied bias voltage of up to 0.1 V (before forming process) show a remarkably low current of ≤100 pA. This value is more than 6 orders of magnitude smaller than the previously reported currents of higher than 100 µA in STO-based memristor TEM *lamella* devices[14], which are contacted using the GIS-deposited

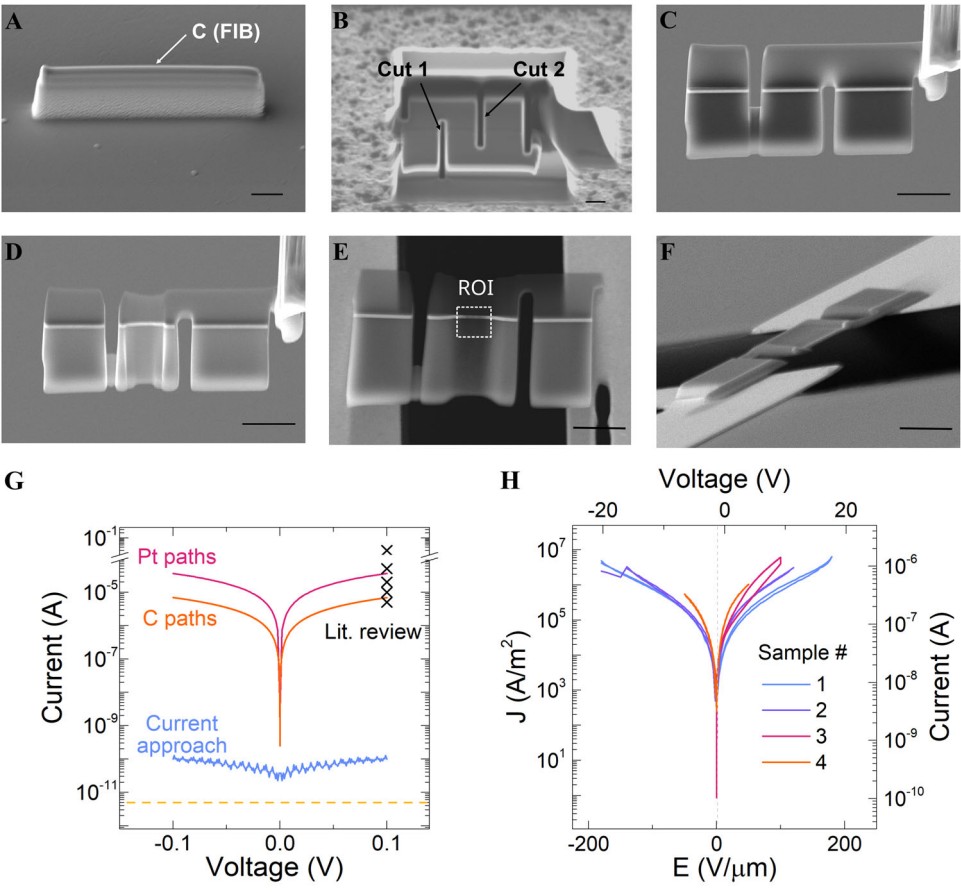

**Fig. 1 Focused ion beam (FIB)-based preparation routine and improvements in the leakage current. A–F** Scanning electron microscope (SEM) images showing the improved FIB sample preparation routine. The scale bar is 2 μm. **A** FIB deposited carbon protection layer (white arrow in **A**). **B** FIB cuts along the two-terminal thin-film heterostructure device to limit the current flow path from the top (cut 2) to the bottom (cut 1) electrodes. **C** Lift-out procedure using a micromanipulator. **D** FIB thinning process of the transmission electron microscopy (TEM) sample. **E** Overview image of the contacted TEM *lamella* device on the MEMS-based chip. The TEM *lamella* device is placed over the chip leads without the use of any FIB deposited material. Note the region of interest (ROI) highlighted with white dashed lines. **F** Side view SEM image of the device. **G** Current-voltage measurements of Pt/STO/Nb:STO TEM *lamella* devices contacted by the use of FIB-deposited Pt (red curve) and C (orange) paths, these result in short-circuit conditions (see also Fig. S3). Similarly, high-current levels have been reported in literature for other two-terminal devices ('X' mark) (see also Fig. S4). In contrast, our here described novel preparation approach (blue curve). The current measurement threshold set by an empty chip is indicated by the yellow dash line. **H** Demonstration of high-reproducibility rate based on the leakage current measurements of four TEM *lamella* devices extracted from the same Pt/STO/Nb:STO macro device.

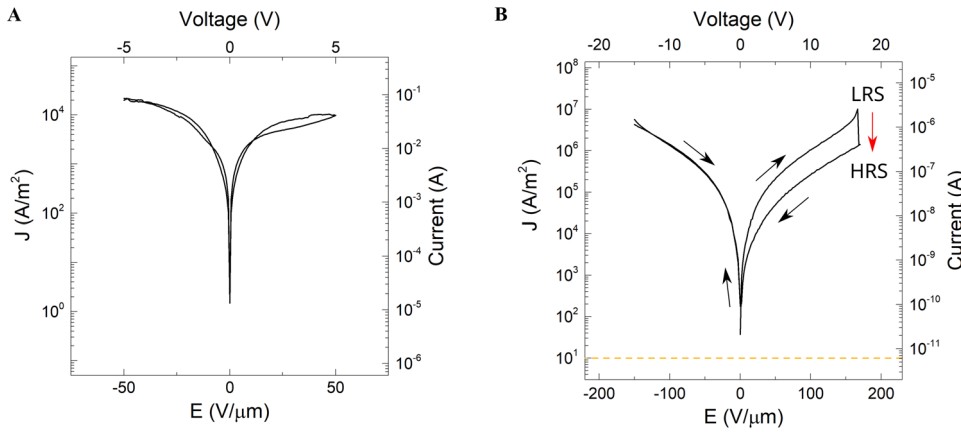

**Fig. 2 Leakage current and current density analysis of a Pt/STO/Nb:STO device. A** Electric response of a millimeter-sized device (area ~4 × 10⁻⁶ m²) in low resistive state (LRS) prior resistive switching (Fig. S13). **B** Electric response of the transmission electron microscopy (TEM) *lamella* device (area ~3 × 10⁻¹³ m²) extracted from the same macro device shown in (**A**). The TEM *lamella* device itself generates, as expected, lower current values than the macro device; therefore, the threshold set by an empty chip (yellow dash line) must be taken into consideration for low leakage devices (Figs. S10 and S12). At 18 V, the TEM *lamella* device could be switched from a LRS to high resistive state (HRS), as indicated by the red arrow, demonstrating memristor capabilities (see Fig. S14).

approach. When measuring at higher bias voltages of up to 5 V ($E$ of up to 50 V/μm in the STO interlayer), currents of ≤10 nA are measured, corresponding to a current density of ≤$10^4$ A/m² (Fig. 2B). Nearly identical values of the current density are observed in the macroscopic memristor (a1) at the same bias voltages (Fig. 2A). These results indicate that the electrical performance of the fabricated TEM *lamella* nanodevices, despite their small dimensions, closely resemble that of the corresponding macroscopic devices. Although there is inherent variation in the electrical properties due to the FIB device lamella fabrication. For instance, as observed in Fig. 2B, when applying the higher voltage of approximately 18 V to the TEM *lamella*, the I–V curve resembles the shape typical of the reset process for the macroscopic memristor (Fig. S13)[26]. Additionally, full memristor operations on TEM *lamellae* of the (a1) and (a2) devices, have been achieved inside a microscope, as shown in Fig. S14.

**Atomic-resolution imaging under *Operando* conditions.** One of the main objectives of in situ TEM biasing experiments is the direct observation of structural changes prompted by an applied *stimulus*. However, the origin of these changes might be ambiguous among short-circuited and operative TEM *lamellae* when using MEMS-based chips. In short-circuited samples, the electric field might be distributed non-uniformly across the *lamella* resulting in non-operative devices (Fig. S5). A short-circuit-free TEM *lamella* device assures that the electric potential is directly applied to one of the two-terminal electrodes of the MIM device. Here, using our short-circuit-free approach we demonstrate a direct correlation between the resistive switching behavior and a noticeable structural change in memristor (a1) (Fig. 3). The formation of extended defects in STO such as dislocations[26], stacking faults[27] or stoichiometry-based phase formation[28] is

expected during resistive switching. However, the nature of these defects and their formation mechanisms are still under debate[29]. Viewing along the [110] zone axis (ZA) in STO, we observe in situ biasing-induced defect formation occurring during the reset process of memristor (a1) while we simultaneously measure its electric response. To facilitate visualization, Fig. 3A, B show models of a pristine and after-reset states of STO viewed along the [001] ZA, respectively. The observed extended defect is composed of consecutive defect slabs (Fig. S15) that are formed in the high resistive state (HRS). The corresponding simulated high-angle annular dark-field (HAADF)-STEM images along the [110] ZA are shown in Fig. 3C, D (se also Fig. S16). The experimental HAADF-STEM images, obtained along the [110] ZA for both, for the pristine or low resistive state (LRS) and the HRS are shown in Fig. 3E, F, respectively (see also Fig. S17). The images were acquired in the exact same region of the STO layer before and after the device reset, and the I–V curve measured during image acquisition is shown in Fig. 2B. Interestingly, a displaced atomic column in each unit cell is evident in the (110) plane that was previously not visible in the pristine or LRS state (Fig. 3F and Fig. S17). This suggests the formation of interleaved extended defect slabs along the [010] direction. The HAADF-STEM simulations fit the experimentally obtained HAADF-STEM images, therefore corroborating the observation.

**Electron beam-induced current in STEM.** Finally, to further investigate the functionality of thin-film oxide-based stack devices inside a TEM, the effect of the electron beam (e-beam) irradiation on sample current was assessed. E-beam irradiation can alter devices during TEM biasing experiments, possibly hampering the operation of a *lamella* device due to beam-induced charging, heating, or oxidation processes[15,30]. An e-beam current threshold

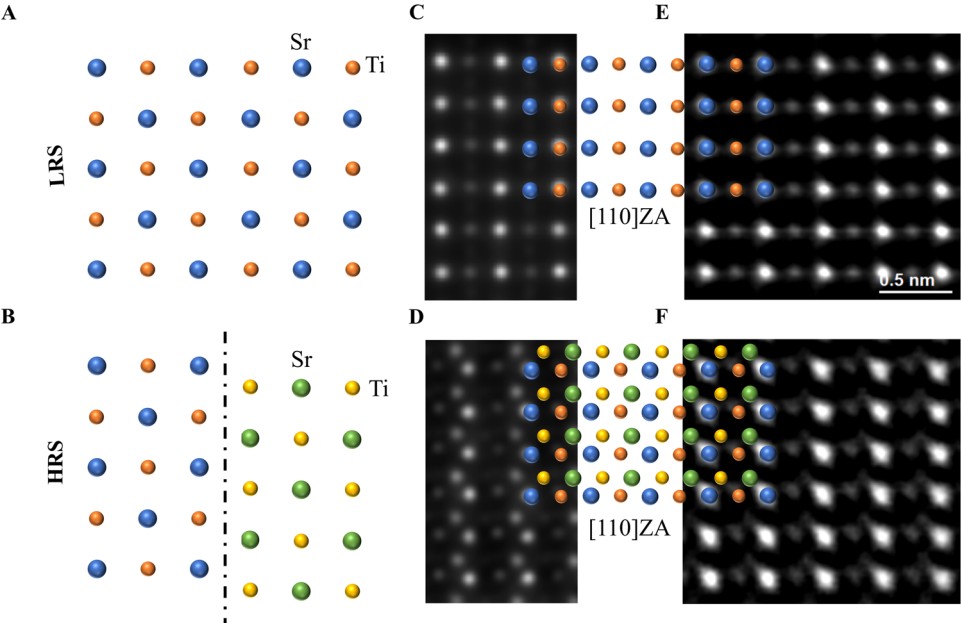

**Fig. 3 In situ biasing scanning transmission electron microscopy (STEM) observation of extended defect formation in a Pt/STO/Nb:STO TEM *lamella* device during resistive switching. A** Schematic of the STO atomic structure in pristine or low resistive state (LRS) state observed along the [001] direction (Sr in blue and Ti in orange). **B** Schematic of defect formation based on consecutive gliding planes formed during the device reset process as viewed along the [001] zone axis (ZA). **C** Simulated STO in pristine or LRS state observed along the [110] ZA. **D** Simulated extended defect observed after device switching to a high resistive state (HRS) and as viewed along the [110] ZA. **E** Experimental high-angle annular dark-field (HAADF)-STEM image of STO acquired along the [110] ZA prior to the reset process. **F** Experimental HAADF-STEM image, at exact same position, viewed along the [110] ZA after reset. The observed extra atomic column (green Sr in the inset model) suggests the formation of extended defects as depicted in the model insets. The images shown in (**E**) and (**F**) were acquired while simultaneously measuring the electric response of the device at both LRS and HRS correspondingly, as shown in Fig. 2B (red arrow).

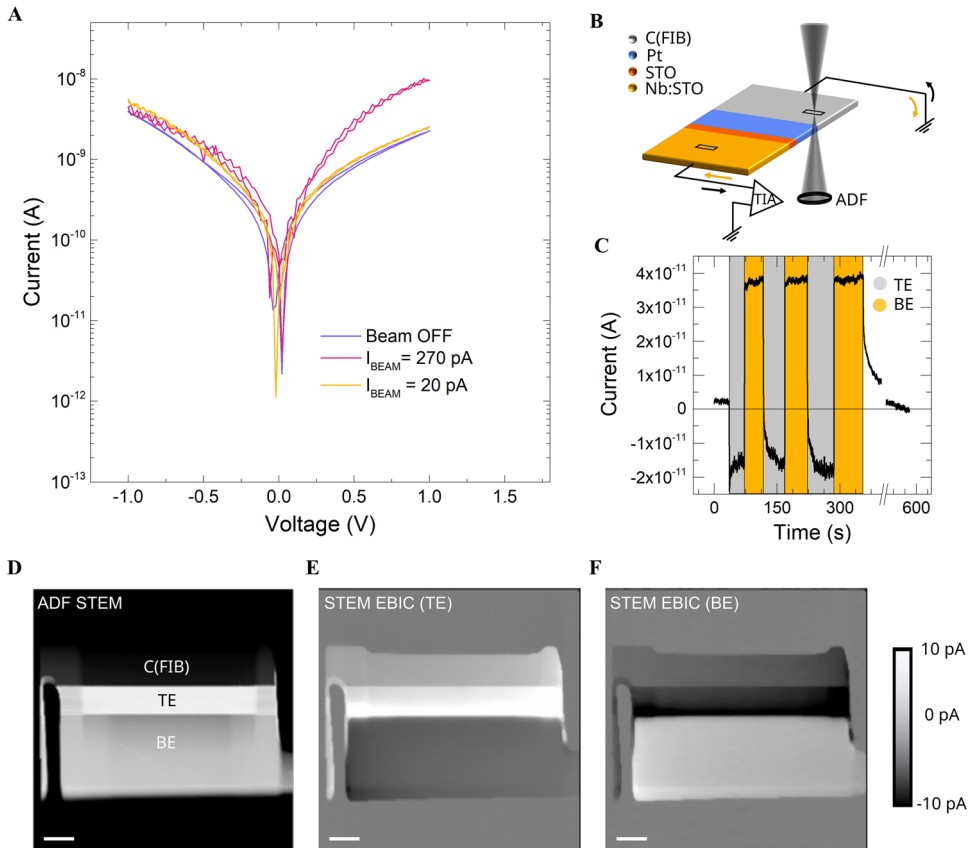

**Fig. 4 Influence of the electron beam on the device (a1) connectivity. A** Effect of electron beam irradiation on the electrical response of the Pt/STO/ Nb:STO TEM *lamella* device. Negligible influence of the e-beam is observed with a e-beam current of ~20 pA (yellow line) when compared with the off beam-state (blue line). A slight change in current in the positive bias regime was measured when the e-beam current was increased to ~270 pA (red line). **B** Sketch of the electron beam induced current (EBIC) produced by the secondary electron emission of a transmission electron microscopy (TEM) *lamella* device. The black and yellow arrows represent the induced current flow direction in the TE and BE, respectively. **C** Single-point EBIC current measurements according to the methodology described in (B) at 0 V. As expected, opposite current signals between the TE (gray zone) and BE (yellow zone) were observed respectively. Note that to find the reference signal response of the device, the initial and final steps during current acquisition were attained with an e-beam in off-state (white zones). **D** Annular dark-field (ADF) STEM of a device along with simultaneously acquired STEM EBIC images from the top (**E**) and bottom (**F**) electrodes. The scale bar is 2 μm.

can be established by testing on/off states (blanking of the e-beam) while simultaneously measuring the electrical response of a device[6,30]. Here, we limit the STEM beam current such that the device remains electronically operative while scanning the probe. To do this we acquired sweep biasing loops of the (a1) memristor device while irradiating the TEM *lamella* at different e-beam currents (Fig. 4A and Fig. S18). When using low e-beam currents (~20 pA) the I–V curve obtained was comparable to the values measured during the beam-off state. Increasing the e-beam current to larger values (~270 pA) produced a slight change in current in the positive bias regime. Though beam effects become obvious in these devices at higher beam currents, in this case >270 pA, the effects are insignificant for e-beam currents significantly below this level.

While beam effects can confound electrical testing in sensitive devices, they can also be used to perform local electronic probing on devices. For example, electron beam-induced current (EBIC) measurements can also be used to verify whether the TEM *lamella* device is properly connected[26,31–33] (Fig. S19), confirming that low leakage values are not simply the result of poor electrical connection. An electron beam will eject secondary electrons (SEs) from a sample, leaving behind holes; this hole current is known as SE emission EBIC, or SEEBIC. SEEBIC measurements can indicate local conductivity in a sample, exhibiting bright (hole)

current when the beam irradiates a region electrically connected to the EBIC amplifier and dark (electron) current when it is incident on a conductor which is disconnected from the amplifier[32,33]. Figure 4B illustrates single-point measurements of this effect using the (a1) device. As indicated by gray and yellow arrows in the sketch, the EBICs in the top (TE) and bottom (BE) electrodes of the TEM *lamella* device must be of opposite polarity if the device is properly electrically contacted, as is here the case. The zero current reference value is determined by briefly blanking the e-beam at the beginning and the end of every image. Hence, as seen in Fig. 4C, the experimental current values obtained within the TE and BE were indeed opposite, demonstrating sample connectivity.

An EBIC image is generated by acquiring current measurements pixel-by-pixel as the beam in scanned in STEM. To confirm connectivity in the device, we generate two STEM EBIC images with separate EBIC amplifiers connected to the top (Fig. 4E) and bottom (Fig. 4F) electrodes, while simultaneously collecting the annular dark field STEM signal (Fig. 4D). Signal in Fig. 4E is bright in the FIB-deposited carbon and TE layers, while the BE alone is bright in Fig. 4F, indicating that the two sides of the device are electrically isolated by the device layer between the TE/BE. These images demonstrate that each device electrode is well-connected to an electrode on the MEMS chip (Fig. S23), but

well-isolated from the opposing device electrode (i.e., no leakage paths across the device), which is precisely the aim of our sample preparation procedure.

The FIB-based preparation methodology for MEMS-based in situ TEM biasing described here allows the proper operation of electrically contacted oxide-based two-terminal stack devices while imaging in the TEM. The elimination of electron or ion beam-induced deposited materials during the FIB TEM *lamellae* preparation process while using MEMS-based chips avoids the formation of stray leakage current paths and enables the realistic electrical operation of two-terminal oxide devices under biasing conditions. A detailed and systematic comparison between current and current densities of millimeter-sized thin-film-based devices and their counterpart TEM *lamellae* is a viable and indispensable way to assure an appropriate electrical contact for further in situ/operando TEM experiments. While we have shown that eliminating GIS- deposited material from the process substantially reduces leakage, other effects, such as surface damage, could also contribute to the remaining leakage and therefore represent a direction for future development. These results open the possibility to establish structure-property correlations while simultaneously measuring realistic changes in current and voltage values.

## Materials and methods

**Macro device fabrication**. The investigated MIM macro devices presented in Table S1 were fabricated as follows:

(a1) The STO-based memristor test structure was fabricated on a $5 \times 5$ mm$^2$ (001) STO substrate doped with Nb at 0.5 w.%. At this doping level, the substrate material is a degenerate semiconductor with a relatively high charge carrier concentration of more than $1 \times 10^{20}$ cm$^{-3}$, resulting in a metallic-like resistivity-temperature dependence[34]. The low room-temperature resistivity below $1 \times 10^{-4}$ $\Omega$m of the 1 at % Nb-doped STO substrates makes them suitable for use as metallic ("M") bottom electrodes for the investigated memristor devices. At first, a 100 nm thick SrTiO$_3$ film was grown using pulsed laser deposition (PLD). The substrate was mounted on an Inconel sample holder and introduced into the UHV chamber at a base pressure of $1 \times 10^{-9}$ Torr. During the STO film growth, the sample holder was heated up to 630 °C from the backside using a high-power near-infrared diode laser while the sample temperature was measured using a pyrometer. A $10 \times 10$ mm$^2$ STO single crystal was used as a PLD target ablated using a KrF excimer laser beam at 248 nm with a fluence of 0.6 J/cm$^2$ and a pulse repetition rate of 2 Hz in an oxygen atmosphere. A 200 nm thick Pt film was grown using DC magnetron sputtering at room temperature.

(b1, b2) Tunable capacitors (varactors) based on tunable dielectric Ba$_{0.5}$Sr$_{0.5}$TiO$_3$ (BST) were fabricated on $5 \times 5$ mm$^2$ (110) GdScO$_3$ substrates. First, a buffer layer of 10-unit cells of STO was deposited onto a GdScO3 substrate in vacuum (at a base pressure of about $3 \times 10^{-8}$ Torr) at 630 °C with a laser fluence of 1 J/cm$^2$ and a pulse repetition rate of 2 Hz. Then, a SrMoO$_3$ film with the thickness of (b1) 5 μm ((b2) 4 μm) was grown at 630 °C in an argon atmosphere at (b1) 30 mTorr and ((b2) 60 mTorr) at a laser fluence of 0.6 J/cm$^2$ and a repetition rate of 20 Hz. In a further step, a 12-unit-cell-thick (b1) BST ((b2) 3 at. % Mn-doped BST) interlayer was deposited at 0.6 J/cm$^2$ in vacuum to prevent oxidation of the underneath SrMoO$_3$ layer. Then, a tunable dielectric layer of (b1) BST ((b2) 3 at. % Mn-doped BST) was grown at 0.6 J/cm$^2$ in an oxygen atmosphere at 35 mTorr at 470 °C. Thus, the total thickness of the (b1) BST ((b2) 3 at. %Mn-doped BST) of (b1) 50 nm ((b2) 100 nm) was grown. Then the samples were cooled down to room temperature in vacuum at a rate of 30 K/min. At the next step, a metallic bilayer top electrode

comprised of 30 nm thick Pt and 400 nm thick Au films were grown on top of the Mn-BST/SrMoO$_3$/SrTiO$_3$/GdScO$_3$ heterostructure (b2) by DC magnetron sputtering. For the BST/SrMoO$_3$/SrTiO$_3$/GdScO$_3$ heterostructure (b1), a 4 μm thick Au film was electroplated on top of the 40 nm thick Pt film grown by DC magnetron sputtering.

At the final step, the Pt film of the Au/Pt bilayers of the (b1) and (b2) BST-based varactor test structures were patterned into a central circular patch with a diameter of $D = 20$–60 μm, surrounded by a concentric ring-shaped ground plane with an outer diameter of $D_{out} = 350$ μm, using a standard photolithographic lift-off process (Fig. S20). Leakage currents of the produced test structures were measured in a ground-signal-ground (GSG) configuration at room temperature, using a vector network analyzer E4991B by Keysight Technologies. DC bias voltage was applied by an external voltage source. For the investigated test structures with the aforementioned dimensions, their leakage current is dominated by that of the small central "Signal" MIM element. Electric properties of similar BST-based varactor test structures at high frequencies up to 3 GHz were reported elsewhere[23–25].

**FIB *lamella* device fabrication**. The FIB system used for device fabrication was a JEOL JIB-4600F with an implemented MM3 Kleindiek micromanipulator. The preparation routine starts with a FIB-deposited carbon protection layer of approx. 2 μm, to prevent Gallium (Ga$^+$) contamination during the subsequent steps. Cutting of trenches at 30 kV with a beam current of 10,000 pA is followed. Then, two parallel undercuts to split the bottom and top electrodes of the device are made at 30 kV and 1000 pA followed by the lift-out process using a micro-manipulator. Various steps of thinning at different current and voltage parameters, as described by Zintler et al.[12], are executed to assure sample quality. It is important to highlight that all the thinning steps, until reaching electron transparency, are performed on the micromanipulator, far away from the MEMS-based chip to avoid any Gallium contamination of the chip. After the thinning process is completed, the TEM sample is placed directly over the leads of the MEMS chip. This attachment step can be performed either exclusively with SEM view or with both SEM and FIB views. No damage attributable to ion irradiation has been observed during this step. However, it is recommended to utilize only the SEM view to mitigate any potential damage. Next, a careful detachment of the micro-manipulator is accomplished by cutting the micromanipulator tip at 10 kV. The TEM *lamella* device is now fixed to the leads solely by van der Waals forces. Optionally, the use of SEM glue could be used to reinforce the *lamella* attachment, no alteration in the electrical response of the TEM *lamella* device with and without glue has been observed (Fig. S6).

**Electrical characterization of a TEM *lamella* device**. All the TEM *lamella* devices were electrically tested using a DENS solutions D9 + TEM holder link to a DENSsolutions interconnect unit and a Keithley 2450 source measure unit. The measurements were always carried out in ultra-high vacuum inside the TEM. Sweep biasing experiments were performed with a step voltage of 0.025 and a delay time of 0.1 s. The potential, in all cases, was applied directly to the top electrode of the TEM *lamella* devices through the conductive Carbon layer used as a protection film during the FIB process. To calculate current density, a fixed area of the TEM *lamella* was established to be $3 \times 10^{-13}$ m$^2$, based on the average length of 3 μm and thickness of 100 nm (Fig. S7 and S21).

**The current threshold of an empty MEMS-chip**. The "Lightning" in-situ MEMS-based TEM sample stage for heating and biasing by DENSsolutions yields a leakage current of about $5 \times 10^{-12}$ A (Fig. S12), which was measured using the experimental setup shown in Fig. S22. This value is a low current limit when measuring the electric properties of high-resistive devices such as the varactor b2 (Fig. S10).

**(S)TEM image parameters**. The atomic resolution HAADF STEM images shown in Fig. 3 were obtained in an Aberration-Corrected JEOL JEM-ARM200F operated at 200 kV, approx. 20 pA e-beam current, and with a semi convergence angle of 25 mrad. The simultaneous annular bright-field (ABF)-STEM images were acquired with an 11 to 23 mrad angle. Lower magnification of the HAADF and ABF-STEM images of Fig. 3. are shown in Fig. S17. It is worth noting that the images prior to and after resistive switching were taken in the same conditions and location, this means, no change in focus or any other microscope parameter was tuned between image acquisition.

**Extended defect model**. The simulation model of STO at LRS was created based on the crystallographic information file (CIF) (icsd-262269). The original unit cell of STO is in Pm-3m space group with lattice constants of 3.905 Å × 3.905 Å × 3.905 Å. The projection vector and upward vector were assigned to be [110] and [1–10] of the original unit cell, respectively. Both pristine and HRS supercells were assigned as the same thickness of 182.25 Å. According to the displacement of atomic columns shown in Fig. 3, consecutive slip planes were created by shifting interval unit cells 75 pm and 150 pm along [00–1] and [1–10], respectively, as illustrated in Fig. S15C. The Ti atoms in the remanent supercell were shifted 20 Å along [1–10].

**Simulation of HAADF-STEM images**. The STEM simulation of the extended defect model was implemented by applying the Multislice algorithm in Prismatic[35]. The simulated probe size (named as pixel size in Prismatic) was 0.05 Å × 0.05 Å. The beam energy was set to 200 KeV. A step size of the e-beam probe for both X and Y directions was set to 0.2 Å. All abbreviation values were assigned to 0 Å. The simulated HAADF-STEM images were obtained by applying virtual detectors of 90 mrad to 125 mrad (Fig. 3C, D and at lower magnification in Fig. S15).

**Electron beam irradiation threshold**. The electrical response shown in Fig. 4A was acquired in STEM mode. All layers of the TEM *lamella* stack device (a1) (Mag: 120k) were scanned while simultaneously acquiring the current values. 512 × 512 images were constantly obtained (live view) with a dwell time of 1 μs, resulting in a time of 0.26 s for each frame. The stack device was scanned horizontal (refer to Fig. S18). The time step for the sweep bias was set to 0.12 s. Thus, the current values in each voltage step are an average obtained for all induced currents of each thin-film layer, consequently, it is not an effect of any specific material as later used in Fig. 4C, but indeed represents the effect of the e-beam on the electrical response of the device.

**Electron Beam Induced Current (EBIC)**. It is known that electron beam interaction with matter in STEM mode generates induced currents that can be used to performed electron beam induced current (EBIC) imaging. The interactions can be classified into three types; electric-field (electron-hole pair generation), absorbed current and SE related. In the devices here investigated, no electron-hole-induced currents are expected (unless defects are present). Absorbed currents, depending on material density, could be considered negligible. Therefore, the main source of

induced currents is related to the holes produced by the emission of SEs, or so-called secondary electron–electron beam induced current (SEEBIC)[32,33]. A sketch of the mentioned induced currents is depicted in Fig. S19.

**Single-point STEM-Electron Beam Induced current measurement (EBIC)**. A Kleindiek EBIC transimpedance amplifier connected in series between the TEM *lamella* device and the source meter unit was installed to amplify the current signal (few picoamps) that was induced locally on the sample. The experiment was carried out at 0 V. Therefore, the curve generated in Fig. 4C was normalized, fitting the minimum acquired current with a zero-reference current.

**SEM-EBIC**. The same EBIC amplifier used in STEM was connected to an SEM/FIB to acquire the images in Fig. S1. The TEM *lamella* device was electrically contacted using a probe shuttle with implemented nanomanipulators.

**STEM EBIC image acquisition**. The 256 × 256-pixel images in Fig. 4D–F and Fig. S23 were acquired simultaneously with an FEI Titan 80-300 TEM operated at 300 kV in STEM mode with a 200 pA beam and a 100 s frame time. The STEM EBIC images were acquired using a two-channel STEM EBIC system from NanoElectronic Imaging (NEI) incorporating a custom biasing holder manufactured by Hummingbird Scientific. No filtering has been applied to the images.

**MEMS-based chips**. The design and development of the MEMS-based chips used in this work have been realized in constant collaboration with DENSsolutions[12,19]. Moreover, the FIB-based sample preparation described in the manuscript has also been employed to the NanoElectronic Imaging (NEI)-based made chips to later proceed with STEM-(SE)EBIC measurements.

## Data availability

All data needed to evaluate the conclusions in the paper are present in the paper and/or the Supplementary Materials. Additional data can be found at TU datalib [https://tudatalib.ulb.tu-darmstadt.de/handle/tudatalib/1892].

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

## Acknowledgements

We thank U. Kunz for technical assistance with the FIB; D. Nasiou and T. Zhang for helpful discussions; S. Matic, H. Maune, and R. Jakoby for electrical measurements of the macro devices (b1 and b2) (TU Darmstadt, Germany); S. Wang for helpful discussions with atom displacement measurements (Beihang University, China); S. Cho, M. Acosta and J. MacManus-Driscoll for providing the macro device (a2) (U. Cambridge, UK); A. Rummel for technical advice with the Kleindiek EBIC amplifier (Kleindiek Nanotechnik, Germany). W.A.H. and B.C.R. are founders of NanoElectronic Imaging, Inc. O.R.-B., T.J., R.W., A.Z. and L.M.-L. acknowledge funding from the European Research Council (ERC) "Horizon 2020" Program under ERC-StG Grant No. 805359-FOXON and ERC-PoC Grant No.957521-STARE. O.R.-B., T.J., R.W., A.Z., Y.R., A.A., L.A., P.K. and L.M.-L acknowledge funding from the German Research Foundation (DFG) - project number 384682067. Y.R., A.A., L.A. and P.K. acknowledge financial support from the DFG project 206658696. The work leading to this publication has been undertaken in the framework of the projects WAKeMeUP under grant agreement no. 783176. This project has received funding from the ECSEL Joint Undertaking (JU) under grant agreement No 101007321. The JU receives support from the European Union's Horizon 2020 research and innovation program and France, Belgium, Czech Republic, Germany, Italy, Sweden, Switzerland, Turkey. Funding by the Federal Ministry of Education and Research (BMBF) under contract 16MEE0154 is gratefully acknowledged. This publication was made possible by the Open Access Monographs Fund of the University and State Library Darmstadt, and by Projekt DEAL.

## Author contributions

Conceptualization: O.R.-B., L.M.-L. Methodology: O.R.-B, T.J., Y.R., A.A., T.V.O., Y.P., W.A.H., B.C.R., H.P.-G., P.K., L.M.-L. Investigation: O.R.-B., T.J., R.W., A.Z., E.A., L.M,-L. Visualization: O.R,-B, T.J. Funding acquisition: P.K., L.A, L.M,-L. Project administration: L.M,-L. Supervision: P.K, L.M,-L. Writing – original draft: O.R,-B. Writing – review & editing: O.R,-B, A.Z, W.A.H, B.C.R., P.K., L.M.-L.

## Funding

## Competing interests

The authors declare no competing interests.
