## [Peer Review File · Communications Engineering]

Reviewers' comments:

Reviewer #1 (Remarks to the Author):

The authors used MEMS-based chips with specific designs to reduce the stray leakage current paths in conventional in-situ TEM electrical biasing. The results are interesting, but the following issues need to be addressed:

1. The authors claim the impact of leakage current path. Would such leakage current path affect the physical mechanism of resistive switching?
2. Fig.3 shows excellent HRTEM images during LRS and HRS. How about the process? What is the oxygen ion's movement or trajectory during the switching process? It is a pity that there were no videos or details of the in situ TEM experiment.
3. The results of EBIC are exciting. What is the spatial resolution of EBIC? Is it possible to observe the phase/grain boundary structure from the EBIC results?
4. I enjoy reading the whole manuscript with new technique improvements for the in situ TEM holders. However, the overall manuscript is not focused. What is the novelty of the work? Are there new findings based on improved tools? Please address the scientific discoveries in the revised manuscript.

Reviewer #2 (Remarks to the Author):

The paper "Operando Two-terminal Devices inside a Transmission Electron Microscope" presents a TEM sample preparation process to strongly limit current leakage during operando experiments and the results obtained on different memresistor devices. In the section on FIB-based preparation, the aim is to avoid Pt and C contamination when creating electrical contacts using FIB-GIS. In the devices section, the electrical responses of FIB-prepared nanodevices (on MEMS-based chips) are compared with the corresponding macroscopic devices. The authors also address the issue of beam-induced current in the STEM technique.

This is a nice manuscript, relevant for TEM operando experiments when biasing excitations are used. My main point is the scope and the organization of the paper: when I agreed to review this article by reading the abstract, I thought I would get more descriptions of the preparation process and the different effects taken into account, and I did not expect to get half the article on device studies, with a relatively in-depth study of the atomic structure before and after biasing with high-resolution imaging simulations. There is only one page on FIB process and electrical responses (macro vs micro). The abstract needs thus to be modified. In addition, there are 29 figures in the supplementary material, which is far too many, with details that would be more interesting in the body of the article, or others for DENs marketing. Particularly, I don't know if figures 22, 24, 25 and 28 are that useful for understanding the article. In the same logic, I don't see the interest of figure 27.

As a consequence, I recommend the publication of this article after reducing the supplementary

text and modifying the abstract to make it more relevant to the rest of the article. Others major points that require attention, in particular more details on the specimen preparation are required prior publication:

- The two-terminal devices are nominated MIM devices in the manuscript. However, the Nb-doped STO is in general commonly regarded as semiconductor, and the doping concentration (heavily or weakly) determines whether the charge carrier level. What is the expected density of carriers and can we consider the Nb-doped STO as a metal?
- For specimen preparation, in order to avoid damage to the MEMS chip, the authors proposed to perform the final thinning of the TEM lamella (<100 nm) directly on the micromanipulator before attaching the lamella to the MEMS chip by van der Waals forces (with/without SEM-compatible glue). Given that the thinning of the lamella is carried out when it is mounted on the micromanipulator, the authors should perhaps address the tilting of the micromanipulator (+1°, 2°, ...) in the JEOL JIB-4600F. Whether the increased tilting angles will create vibration problems or not during thinning, and thus ensure a homogeneous thickness of the lame?
- How is the final lamella carefully transferred to the MEMS chip? As the micromanipulator operates in 3D space, two detector views (SEM+FIB) are generally required. Does this mean that this process is carried out only by SEM view, without any FIB view? It is not clear.
- How do authors protect the thinned devices from ESD (electrostatic discharges)? Are ESD critical for the thinned devices? Have they already observed such discharges when mounting the chip on the biasing holder, or when depositing the lamella on the chip, with an electron or ion beam effect?
- How is the glue deposited? Detail on the glue would be interesting for the readers.
- After transfer, the lamella is carefully detached by cutting the micromanipulator tip at 10kV. Do you think this essential procedure will create Ga contamination for the chip?
- What is the final cleaning voltage for the lamella? With additional Ar milling or not?
- In addition, two undercuts are made (at 30 kV and 1000 pA) to separate the top and bottom electrodes of the device prior to lifting, in which case the FIB column is tilted relative to the lamina (probably 54° or 52°). In the reviewer's opinion, this method of cutting can create an uneven distribution of lamina size between the front and back views. It is preferable to perform the undercuts perpendicularly.
- I regret that there is not a more in-depth analysis of the effect of amorphisation and Ga contamination of the lamella surfaces during FIB thinning on the electrical measurements. It would be interesting to carry out a cross-sectional observation of the prepared lamella in order

to measure the thickness of the surface modified by the ion beam. This observation would be an important added value to the paper.

For this routine to be feasible for users and an important part of the process, these issues should be clearly stated in the device manufacturing section.

- Whether the resistive switching is voltage driven in (a1)? Considering Fig.2(A), Fig. 13 Sup, Fig.14 Sup (macro- or nano-, inside or outside the microscope), is it bipolar resistive switching or uni-polar for (a1)?

- If the authors keep the part on the in-depth study of the atomic structure, they need to discuss in brief the switching mechanism with I-V curves and possible structural variation. It is interesting that extended defect formation was observed during resistive switching. As a comment, the element trace signal from HAADF-STEM is a little weak in Fig. 3 and iDPC may improve (with light O atoms or O vacancies). Is, is the switching related to a possible schottky barrier modulation since we could consider a schottky junction at the Pt/SrTiO₃ interface and the n-n⁺ junction at the SrTiO₃/Nb:SrTiO₃ interface?

Minor points :

(1) "Lamella" or "Lamellae", this should be uniform in the text.

(2) Fig.1 Supplementary: While Pt deposition using GIS-FIB or GIS-SEM will create surface contamination, the influence is controlled by the used parameters during deposition process. The authors should give the parameters of deposition as a reference for the readers.

(3) Fig.6 Supplementary: Electric response of a TEM lamella device with glue in black is missing, or is it totally overlapped with the red curve? Have the curves been obtained on the same sample (glue deposited after a first electrical measurement)?

(4) Fig.15 Supplementary: The text of the coordinate system is too small and the scale bars for the STEM images are missing.

(5) Fig.23 Supplementary: Here is a good example to show the cases with or without lateral cuts. However, the text in the multi-figures are unreadable. Need to be reproduced. Where is the "dashed black line" described in the legend (line 7, p44)?

(6) P32 Fig.11 Supplementary: The authors should note a1/a2/b1/b2 in the figure captions, since these stacking figures are complementary to Table 1.

Reviewer #3 (Remarks to the Author):

The authors present an interesting study demonstrating a new approach to producing lamella devices for operando TEM study. Rather than GIS deposition to address the electrodes of the lamella, FIB cutting is used to introduce isolation of the electrodes prior to MEMS mounting. This produces a significantly reduced leakage current between the electrodes, with isolation of the electrodes also demonstrated via EBIC. As a case study, this preparation method allows the authors to observe atomic scale structural changes in their device during resistance switching.

The authors also suggest that this paves the way for preparation of devices which more accurately reproduce macroscale behaviour.

I believe that the work shown in this manuscript is novel and valuable to the community, pending answers to a number of comments/queries I have detailed below. In general, I think it is reasonably well-written and that the conclusions are supported by the data. In addition to some minor comments, there are several discussion points which I do not feel are fully supported by evidence, and others which require clarification on exactly how measurements/simulations were carried out. So, I do not feel that any additional experimental work is necessary, but some well-considered changes to the manuscript certainly are in order to make it suitable for publication.

One concern is the quantity of supplementary material which is presented. There seems to be more supplementary content than main manuscript. Is it necessary for so much supplementary information to be present, or could the authors use more citations, for example?

P15 L17 – high vacuum or actually UHV used?

Fig. 1D – is this data for Pt and C the same as in Sup. Fig. 3? Was this device fabricated in the same way as a1 (aside from GIS)? The fabrication details for all samples relevant to the work should really be included. It is also important for the reader to understand exactly how the poor-performance/shorted devices were formed, to be clear that the process is standard GIS (i.e., there isn't an extreme excess of Pt or C deposited to strongly encourage shorted devices).

Fig. 2 – the “switching” here at 18 V seems like an extremely high field, what voltage/polarity is required to switch back to the LRS? I would also disagree with the final sentence in the caption “Note the similar current density values of the TEM lamella device corresponding to the macro device during operation inside a TEM” – the current density in the lamella device at +/- 5 V is around 3×10^5 A/m². In the macro device the current density at +/- 5 V is around 10^4 A/m², which is significantly lower (more than an order of magnitude). In fact, the macro device is closer in current density to the HRS of the lamella.

P3 L42 – assuming the authors are referring to the LRS of the lamella, the current at 5 V is more than 10 nA (closer to 100 nA) and the current density is more than 10^4 A/m² (closer to 3×10^5 A/m²), so I do not agree that it is then reasonable to say on P4 L1 that lamella devices “despite their small dimensions, fully reproduce that of the corresponding macroscopic devices”. Perhaps the authors could reword this section to more accurately describe what the data are showing and indicating about their devices.

Also sup. Fig. 8 caption makes a similar point as above on similarity between lamella and macro devices, but the plots show quite different behavior (different curve shapes). There is also not a

clear set/reset occurring, there is only one distinct jump between states, under positive bias.

The resistance switching of the lamella device e.g. sup fig 14 actually looks better than the macro device e.g., sup fig 13, which looks like it is showing a probe contact/charging artefact (HRS min current not at 0 V). Additionally, the switching to LRS is at a high voltage, so possibly this is just hard breakdown (no reset shown)? Can the authors comment on why their lamella device shows better switching behavior than the macro devices? No info is given on how the macro devices were operated electrically so it's possible there was an issue with the instrumentation. Was the switching of the macro devices done between the S and G locations as depicted in Sup. Fig. 20?

Why are a2, b1 and b2 included in the table/fabrication details? They are not mentioned anywhere in the manuscript, only in the supplementary information. Is it necessary to include them in the table or supplementary information if they are not relevant to the main study and its conclusions?

Following the above comment, it would be good to gauge the repeatability of the process. Is all the data from a single device/fabrication (i.e., sample a1) or the result of multiple fabrications? How widely applicable a solution is the authors' method to addressing the shortfalls of standard GIS lamella preparation? Is it significantly more difficult to get the FIB cuts correct vs doing GIS deposition? (Of course, I appreciate the huge challenge in preparing even a single lamella device!).

P4 L10 – “In short-circuited samples, the electric field is distributed discontinuously across the lamella resulting in non-operative devices (Supplementary Fig. 5)” Where is the evidence supporting this? Sup. Fig. 5 just shows a device with some damage after the application of a relatively high current. In what way is this indicative of a discontinuous field distribution, and has this distribution been measured or simulated in any way? The figure also shows a maximum field of 20 V/um, which for an a1 device (105 nm thick active layer) implies a maximum voltage of 190 V (which is far beyond any normal operational voltage for a resistance switching device), not 2 V, as is shown on the top x axis. Which scale is correct?

How can the authors be sure that there is no drift between the images in Fig. 3E and F?

P5 L35 – “A detailed and systematic comparison between current and current densities of millimeter-sized thin-film-based devices and their counterpart TEM lamellae is a viable and indispensable way to assure an appropriate electrical contact for further in situ/operando TEM experiments.”

P18 L25 – how was the CASINO simulation carried out? There needs to be some more detail in order to support the statement that “Absorbed currents, depending on material density, could

be considered negligible.”

Sup. Fig. 28 – what is the purpose of this Figure? It doesn't seem to be relevant to the manuscript and just shows potential chips for other applications, without any discussion. Is the purpose here to indicate that various GIS-free chip arrangements are possible, in which case are further experiments/figures/discussion required to demonstrate the flexibility of the lamella fabrication process?

Reviewer #1 (Remarks to the Author):

We thank the reviewer for the positive, insightful, and very helpful comments to further improve the manuscript. We hope that all your inquiries are answered satisfactorily.

1. The authors claim the impact of leakage current path. Would such leakage current path affect the physical mechanism of resistive switching?

Indeed, the presence of leakage current pathways formed by Pt or C contamination directs the electrical current along these contaminated regions. Regardless of the applied voltage, if the current does not flow through the dielectric layer of the TEM lamella device, resistive switching in memristors cannot occur. For example, as shown in Supplementary Figure 5, a memristor TEM lamella device in a short-circuit state experiences very high currents (0.02 A) at relatively low voltages (2 V) due to Pt contamination, which leads to localized melting points on the TEM lamella. However, switching did not occur before layer damaged. Furthermore, increasing the voltage will raise the current levels without leading into a change in resistive state but potentially resulting in the failure of the sample and the MEMS chip itself, as shown in the SEM image below.

A note has been added in the manuscript to highlight this important suggestion from the reviewer:

The change to the manuscript text is shown below:

Page 2 lines 46-47: “..., affecting not only the electric responses of MIM devices but also”

2. Fig.3 shows excellent HRTEM images during LRS and HRS. How about the process? What is the oxygen ion's movement or trajectory during the switching process? It is a pity that there were no videos or details of the in situ TEM experiment.

Atomic scale imaging is possible due to the advantages of aberration corrected TEM microscopes in scanning mode (STEM). However, the fastest acquisition time allowed in STEM mode is approx. 1μs per pixel (0.3 s in a 512x512 frame), using standard CCD detectors the resistive switching process is unfeasible to record because it is believed that the structural change occurs within nanoseconds [1]. In this regard, the primary result of this experiment is the observation of the structural changes before and after the switching process happened within a static region.

The images presented in Figure 3 are obtained using HAADF-STEM, which is sensitive to Z-contrast. Oxygen, being of lower Z number, remains invisible, as demonstrated by the simulated HAADF image

in Figure S15. Conversely, Figure S17 displays the corresponding ABF image where oxygen becomes visible. In this instance, no structural changes were detected in the oxygen component. Additionally, the [110] STO orientation in STO is not favorable for observing oxygen cations in perovskite materials, however, this orientation allowed us to clearly observe the formed interleaved extended defect slabs along the [010] direction. All in all, it is planned in the future to use iDPC methods together with in situ TEM experiments to track oxygen movement.

3. The results of EBIC are exciting. What is the spatial resolution of EBIC? Is it possible to observe the phase/grain boundary structure from the EBIC results?

We appreciate the reviewer's acknowledgment of the significance of the EBIC technique. As noticed by the reviewer, the underlying concept of this manuscript is indeed to explore the future potential of observing phase/grain boundary structure formation, akin to the findings demonstrated by our co-authors, Dr. W. A. Hubbard and Prof. B.C. Regan, in their 2022 publication [2], wherein electric breakdown was clearly captured using EBIC techniques. It's worth noting that they employed self-made model memristor systems. In collaboration with Prof. Regan and Dr. Hubbard, our objective is to replicate these observations within actual MIM operative devices, as showcased in this manuscript.

The spatial resolution of STEM SEEBIC has previously been established by Prof. Regan as 2 angstroms in his 2018 article [3]. By using aberration corrected instrumentation this can be further improved. Therefore, we are optimistic that capturing exciting current-driven mechanisms at high spatial resolutions within operational MIM devices will soon be feasible.

4. I enjoy reading the whole manuscript with new technique improvements for the in situ TEM holders. However, the overall manuscript is not focused. What is the novelty of the work? Are there new findings based on improved tools? Please address the scientific discoveries in the revised manuscript.

Thank you for your thoughtful review and valuable insights.

We are addressing a crucial challenge within the community, centered on acquiring devices operating under genuine operative conditions (specifically on two-terminal devices) using MEMS chips in a consistent and reproducible manner, while achieving current levels in the pA regime. The availability of low-leakage TEM devices would revolutionize routine in situ biasing and facilitate broader applications in multi-stimuli experiments.

After dedicated collaborative efforts over recent years, it became evident that utilizing GIS-based methods results in undesired stray leakage current paths that can obscure the genuine electrical response of the device. Consequently, we arrived at the conclusion that to ensure operando conditions for two-terminal devices in in situ TEM applications, a comprehensive approach is essential. This involves meticulous electrical characterization of both macro- and TEM lamella devices, combined with the complete removal of GIS-deposited material.

In addition to these efforts, we introduced the innovative and powerful technique of spatially resolved STEM-(SE)EBIC. This technique provides definitive evidence of sample connectivity, thereby confirming true operability. Notably, we were thrilled to discover that the low-current levels measured through STEM-EBIC area scans closely align with the IV-curves obtained from the same electron-transparent TEM-lamella device. This groundbreaking achievement represents a significant advancement in the field. Furthermore, the outcomes of the STEM-(SE)EBIC analysis challenge certain assumptions, demonstrating that by carefully selecting appropriate FIB-preparation parameters and contacting routines, it is indeed feasible to preserve the nominally insulating STO layer intact.

In order to highlight the findings of this contribution, please note:

Page 3 lines 5-9: “....., In this study, we present evidence of the operability of MIM devices utilizing an innovative FIB sample preparation technique employing MEMS-based chips^{12,19,20}. Our FIB methodology eliminates the occurrence of undesired short-circuits, allowing reliable electric contacting and realistic I-V measurements of *lamellae* of electrode-based thin-film oxide electronic devices inside a TEM”

Page 3 lines 22-24: “....., The obtained low current values fit with the currents expected for a MIM device scaled to TEM *lamella* dimensions (Supplementary Fig. 7), allowing for reliable *operando* TEM studies”

Page 5 lines 25-28: “....., These images demonstrate that each device electrode is well-connected to an electrode on the MEMS chip (Supplementary Fig. 23), but well-isolated from the opposing device electrode (*i.e.*, no leakage paths across the device), which is precisely the aim of our sample preparation procedure.”

References.

- [1] D. S. Jeong *et al.*, “Emerging memories: resistive switching mechanisms and current status,” *Rep. Prog. Phys.*, vol. 75, no. 7, p. 076502, Jul. 2012, doi: 10.1088/0034-4885/75/7/076502.
- [2] W. A. Hubbard, J. J. Lodico, H. L. Chan, M. Mecklenburg, and B. C. Regan, “Imaging Dielectric Breakdown in Valence Change Memory,” *Adv. Funct. Mater.*, vol. 32, no. 2, p. 2102313, Jan. 2022, doi: 10.1002/adfm.202102313.
- [3] W. A. Hubbard, M. Mecklenburg, H. L. Chan, and B. C. Regan, “STEM Imaging with Beam-Induced Hole and Secondary Electron Currents,” *Phys. Rev. Appl.*, vol. 10, no. 4, p. 044066, Oct. 2018, doi: 10.1103/PhysRevApplied.10.044066.

Reviewer #2 (Remarks to the Author):

Many thanks to the reviewer for the very insightful comments and helpful suggestions to improve the manuscript.

The paper "Operando Two-terminal Devices inside a Transmission Electron Microscope" presents a TEM sample preparation process to strongly limit current leakage during operando experiments and the results obtained on different memresistor devices. In the section on FIB-based preparation, the aim is to avoid Pt and C contamination when creating electrical contacts using FIB-GIS. In the devices section, the electrical responses of FIB-prepared nanodevices (on MEMS-based chips) are compared with the corresponding macroscopic devices. The authors also address the issue of beam-induced current in the STEM technique.

This is a nice manuscript, relevant for TEM operando experiments when biasing excitations are used. My main point is the scope and the organization of the paper: when I agreed to review this article by reading the abstract, I thought I would get more descriptions of the preparation process and the different effects taken into account, and I did not expect to get half the article on device studies, with a relatively in-depth study of the atomic structure before and after biasing with high-resolution imaging simulations. There is only one page on FIB process and electrical responses (macro vs micro). The abstract needs thus to be modified. In addition, there are 29 figures in the supplementary material, which is far too many, with details that would be more interesting in the body of the article, or others for DENs marketing. Particularly, I don't know if figures 22, 24, 25 and 28 are that useful for understanding the article. In the same logic, I don't see the interest of figure 27.

As a consequence, I recommend the publication of this article after reducing the supplementary text and modifying the abstract to make it more relevant to the rest of the article. Others major points that require attention, in particular more details on the specimen preparation are required prior publication:

We acknowledge and value the reviewer's perspective on emphasizing the intricacies of the sample preparation method described here. Our article is centered around electronic device properties, sub-nanometer structural alterations, and the application of the STEM EBIC technique to validate sample connectivity.

However, we greatly appreciate the reviewers comment and take his suggestion of modifying the abstract.

The changes to the abstract text are shown below:

Page 1 lines 24-26: "...., Here we demonstrate connectivity of TEM lamella devices obtained from a novel electrical contacting method."

We also thank the reviewer for suggesting a reduction in the number of supplementary images. In the revised manuscript, we have shortened it from 29 to 23 images.

- The two-terminal devices are nominated MIM devices in the manuscript. However, the Nb-doped STO is in general commonly regarded as semiconductor, and the doping concentration (heavily or weakly) determines whether the charge carrier level. What is the expected density of carriers and can we consider the Nb-doped STO as a metal?

The used 1 at.% Nb-doped STO is a degenerate semiconductor which exhibits a relatively high charge carrier concentration above $1 \times 10^{20} \text{cm}^{-3}$. As it was previously shown by Z.Zhang et al. [1], the resistivity of Nb-doped STO single crystals at this rather high doping level increases with temperature and resembles a resistivity-temperature dependence of a metallic conductor rather than that of a “classical” semiconductor. For this reason, we used the Nb-doped STO substrate as a conducting bottom electrode denoted with “M”. We added the corresponding explanation in the “Materials and Methods. Macro device fabrication” section of the manuscript. In the re-submitted manuscript, the Nb doping level of the substrate is given as a weight concentration (0.5 w.%) according to the label of the substrate manufacturer (Crystec).

Please note the changes made in the Materials and methods section in page 17 of the manuscript (lines 4-9):

“..., The STO-based memristor test structure was fabricated on a $5 \times 5 \text{ mm}^2$ (001) STO substrate doped with Nb at 0.5 w.%. At this doping level, the substrate material is a degenerate semiconductor with a relatively high charge carrier concentration of more than $1 \times 10^{20} \text{cm}^{-3}$, resulting in a metallic-like resistivity-temperature dependence [36]. The low room-temperature resistivity below $1 \times 10^{-4} \Omega \text{m}$ of the 1 at % Nb-doped STO substrates makes them suitable for use as metallic (“M”) bottom electrodes for the investigated memristor devices.”

- For specimen preparation, in order to avoid damage to the MEMS chip, the authors proposed to perform the final thinning of the TEM lamella (<100 nm) directly on the micromanipulator before attaching the lamella to the MEMS chip by van der Waals forces (with/without SEM-compatible glue). Given that the thinning of the lamella is carried out when it is mounted on the micromanipulator, the authors should perhaps address the tilting of the micromanipulator (+1°, 2°, ...) in the JEOL JIB-4600F. Whether the increased tilting angles will create vibration problems or not during thinning, and thus ensure a homogeneous thickness of the lamella?

To address your inquiry regarding the final polishing of the TEM lamella, the specific tilting angles will vary based on the material, typically ranging between -7° to +7°. Notably, we haven't encountered vibrations during the thinning process, resulting in highly uniform thicknesses in the region of interest (ROI) after final polishing.

Furthermore, recent advancements in our approach have enabled us to conduct the ultimate polishing step directly on the chip itself, in a free-standing configuration. This entails detaching the manipulator from the TEM lamella, allowing the lamella to maintain a specific angle. This angle facilitates the final polishing of both the back and front sides of the sample through the gap between the electrodes (within the empty space). Consequently, no redeposition occurs, yielding a clean ROI. Following this free-standing polishing step, the sample is manipulated by the micromanipulator, pushing it against the chip's leads to establish electrical contact through van der Waals forces.

While the manuscript outlines the conventional polishing process using the manipulator (which remains entirely valid), the refinement involving the final polishing step in a free-standing position introduces an alternative approach. This innovation not only facilitates the attainment of the intended final thickness but also assists in potential contamination removal if required. However, it's important to note that this particular step presents a challenge that could potentially result in the loss of samples.

- How is the final lamella carefully transferred to the MEMS chip? As the micromanipulator operates in 3D space, two detector views (SEM+FIB) are generally required. Does this mean that this process is carried out only by SEM view, without any FIB view? It is not clear.

In our experience, attaching the TEM lamella can be accomplished solely through the SEM view, provided the correct eucentric height has been determined beforehand. However, utilizing the FIB view does not impact the final outcome and can accelerate the attachment process for the user.

Please note the included lines (12-15) in page 18:

“..., This attachment step can be performed either exclusively with SEM view or with both SEM and FIB views. No damage attributable to ion irradiation has been observed during this step. However, it is recommended to utilize only the SEM view to mitigate any potential damage.”

- How do authors protect the thinned devices from ESD (electrostatic discharges)? Are ESD critical for the thinned devices? Have they already observed such discharges when mounting the chip on the biasing holder, or when depositing the lamella on the chip, with an electron or ion beam effect?

Notably, we've noted a decrease in TEM lamellae affected by electrostatic discharge (ESD) when compared to samples attached using Pt or C paths. In the past, while employing Pt or C paths, we frequently encountered ESD challenges. This concern prompted us to seek alternative approaches to mitigate undesired discharges. Presently, we remain cautious, ensuring proper grounding of both the user and the TEM holder during sample manipulation; however, by using our current sample preparation approach instances of discharge problems are now infrequent.

- How is the glue deposited? Detail on the glue would be interesting for the readers.

The use of SEM glue demonstrates satisfactory electrical properties; we have observed consistent electrical responses both with and without SEM glue. However, it's important to note that the glue could introduce carbon contamination during TEM observation. Consequently, we have opted to stop the use of glue. However, the glue application process involves placing a small quantity within the SEM holder, adjacent to the MEMS chip. Once the TEM lamella is attached through van der Waals forces, we collect a small amount of glue using the micromanipulator tip and then approach the TEM lamella. A minute quantity of glue is then deposited onto a corner of the TEM lamella, following which the manipulator is retracted. Subsequently, the glue is solidified by briefly exposing it to a high ion beam current.

- After transfer, the lamella is carefully detached by cutting the micromanipulator tip at 10kV. Do you think this essential procedure will create Ga contamination for the chip?

Approx. 1 μm region in one of the chip leads might be affected by Ga^+ contamination. However, this damage might not alter the conductivity of the leads. On the other hand, elemental analysis has been conducted on multiple occasions, yet no traces of Ga^+ have been detected, apart from within the C or Pt protection layer, which is commonly observed in TEM lamellae prepared using Ga^+ FIBs. Consequently, there is no additional Ga^+ contamination evident as shown in the image below. It's important to highlight that the extent of Ga^+ contamination can vary significantly depending on the material. In the materials under examination in this manuscript, Ga^+ contamination remains at a minimum.

- What is the final cleaning voltage for the lamella? With additional Ar milling or not?

The final polishing is performed at 5kV, and no Ar milling has been done. Please, refer to Zintler et al., [2] for details in the thinning process.

- In addition, two undercuts are made (at 30 kV and 1000 pA) to separate the top and bottom electrodes of the device prior to lifting, in which case the FIB column is tilted relative to the lamina (probably 54° or 52°). In the reviewer's opinion, this method of cutting can create an uneven distribution of lamina size between the front and back views. It is preferable to perform the undercuts perpendicularly.

We concur with the reviewer's perspective regarding the preference for perpendicular cuts to the TEM lamella facets. Regrettably, due to inherent geometric constraints, achieving a perfectly perpendicular cut in our FIB system proves unfeasible. Instead, we perform an undercut at the maximal allowable angle within our system, corresponding to -5° of stage tilting (or 58° with respect to the FIB column). There have been instances when we attempted undercuts from both facets of the TEM lamella (0° stage rotation and 180° stage rotation) to approach perpendicularity. However, such attempts yielded no discernible alteration in the electrical response between TEM lamella devices cut at true perpendicular or with an angle. As a result, to optimize efficiency, we execute the undercuts at the angle permissible by our system, exclusively from one side of the TEM lamella, as elaborated in the manuscript.

- I regret that there is not a more in-depth analysis of the effect of amorphisation and Ga contamination of the lamella surfaces during FIB thinning on the electrical measurements. It would be interesting to carry out a cross-sectional observation of the prepared lamella in order to measure the thickness of the surface modified by the ion beam. This observation would be an important added value to the paper.

For this routine to be feasible for users and an important part of the process, these issues should be clearly stated in the device manufacturing section.

The thinning process adheres to standard protocols, encompassing rough, medium, and light thinning, followed by polishing stages at low ion beam voltages and currents [2]. The distinctive aspect of this sample preparation technique is its attachment to the chip, which eliminates the need for C or Pt deposition. Consequently, we anticipate that amorphization and Ga⁺ contamination will mirror what's observed in other FIB-based sample preparation routines, whether for MEMS chips or TEM grids. Our experience with (S)TEM observations, both at high and low spatial resolutions, for the same materials examined in this paper but using TEM lamellae prepared within TEM grids, has revealed no discernible disparities compared to TEM lamellae intended for in situ TEM. The same defects and potential contamination are identifiable in both scenarios.

- Whether the resistive switching is voltage driven in (a1)? Considering Fig.2(A), Fig. 13 Sup, Fig.14 Sup (macro- or nano-, inside or outside the microscope), is it bipolar resistive switching or uni-polar for (a1)?

The devices were all electrically tested at ultra-high vacuum within the TEM. The device a1 shows bipolar resistive switching. The in-situ experiment in Fig. 2 shows only one reset step while sup. Fig. 14 A shows forming process and cycling of the same device.

- If the authors keep the part on the in-depth study of the atomic structure, they need to discuss in brief the switching mechanism with I-V curves and possible structural variation. It is interesting that extended defect formation was observed during resistive switching. As a comment, the element trace signal from HAADF-STEM is a little weak in Fig. 3 and iDPC may improve (with light O atoms or O vacancies). Is, is the switching related to a possible schottky barrier modulation since we could consider a schottky junction at the Pt/SrTiO₃ interface and the n-n⁺ junction at the SrTiO₃/Nb:SrTiO₃ interface?

The central emphasis of the article lies not in delving into the resistive switching mechanism in STO-based memristors, but rather in assessing the device's operability within a TEM setting. In order to affirm the device's accurate functionality, we opted for a widely recognized material with established resistive switching mechanisms. Given the multitude of reports discussing extended defects generated during switching, authored by others [3]–[10], our aim was to identify these defects utilizing our unique approach involving in situ TEM biasing sample preparation. It is important to note that the role of these defects in the conduction mechanism of memristors falls outside the scope of the present manuscript. Nevertheless, our future aim is indeed to study the function of these defects in memristor properties.

Minor points :

(1) "Lamella" or "Lamellae", this should be uniform in the text.

In the context of this manuscript we have used the term "Lamella" to refer the singular form, and "lamellae" to refer the plural form.

(2) Fig.1 Supplementary: While Pt deposition using GIS-FIB or GIS-SEM will create surface contamination, the influence is controlled by the used parameters during deposition process. The authors should give the parameters of deposition as a reference for the readers.

Please refer to Zintler *et al.*, [2] the paper covers all the details concerning sample prep for in situ TEM biasing using Pt paths.

(3) Fig.6 Supplementary: Electric response of a TEM lamella device with glue in black is missing, or is it totally overlapped with the red curve? Have the curves been obtained on the same sample (glue deposited after a first electrical measurement)?

For simplicity both curves were plotted in the same plot. However, we thank the reviewer for the suggestion, in the revised manuscript the line thickness has been edited for easy distinction.

It is important to clarify that both curves were obtained from the same sample. Initially, we conducted electrical testing on the sample without any glue. Subsequently, using the same sample, we introduced a quantity of glue and repeated the electrical testing.

(4) Fig.15 Supplementary: The text of the coordinate system is too small and the scale bars for the STEM images are missing.

Thank you, the text in the coordinate system is now larger and the scale bars are included.

(5) Fig.23 Supplementary: Here is a good example to show the cases with or without lateral cuts. However, the text in the multi-figures are unreadable. Need to be reproduced. Where is the "dashed black line" described in the legend (line 7, p44)?

Thanks for noticing these mistakes, the revised version has now been edited.

(6) P32 Fig.11 Supplementary: The authors should note a1/a2/b1/b2 in the figure captions, since these stacking figures are complementary to Table 1.

Thank you, the sample legends are now included in the figures.

References

[1] Z. Zhang *et al.*, "Manipulating the carrier concentration and phase transition via Nb content in SrTiO₃," *Sci. Rep.*, vol. 12, no. 1, p. 2499, Feb. 2022, doi: 10.1038/s41598-021-03199-7.

[2] A. Zintler *et al.*, "FIB based fabrication of an operative Pt/HfO₂/TiN device for resistive switching inside a transmission electron microscope," *Ultramicroscopy*, vol. 181, pp. 144–149, Oct. 2017, doi: 10.1016/j.ultramic.2017.04.008.

[3] R. Waser, R. Dittmann, G. Staikov, and K. Szot, "Redox-Based Resistive Switching Memories - Nanoionic Mechanisms, Prospects, and Challenges," *Adv. Mater.*, vol. 21, no. 25–26, pp. 2632–2663, Jul. 2009, doi: 10.1002/adma.200900375.

- [4] H. Du, C.-L. Jia, J. Mayer, J. Barthel, C. Lenser, and R. Dittmann, "Atomic Structure of Antiphase Nanodomains in Fe-Doped SrTiO₃ Films," *Adv. Funct. Mater.*, vol. 25, no. 40, pp. 6369–6373, Oct. 2015, doi: 10.1002/adfm.201500852.
- [5] C. Lenser *et al.*, "Formation and Movement of Cationic Defects During Forming and Resistive Switching in SrTiO₃ Thin Film Devices," *Adv. Funct. Mater.*, vol. 25, no. 40, pp. 6360–6368, Oct. 2015, doi: 10.1002/adfm.201500851.
- [6] C. Baeumer *et al.*, "Spectromicroscopic insights for rational design of redox-based memristive devices," *Nat. Commun.*, vol. 6, no. 1, p. 8610, Dec. 2015, doi: 10.1038/ncomms9610.
- [7] H. Du *et al.*, "Nanosized Conducting Filaments Formed by Atomic-Scale Defects in Redox-Based Resistive Switching Memories," *Chem. Mater.*, vol. 29, no. 7, pp. 3164–3173, Apr. 2017, doi: 10.1021/acs.chemmater.7b00220.
- [8] D. Cooper *et al.*, "Anomalous Resistance Hysteresis in Oxide ReRAM: Oxygen Evolution and Reincorporation Revealed by In Situ TEM," *Adv. Mater.*, vol. 29, no. 23, p. 1700212, Jun. 2017, doi: 10.1002/adma.201700212.
- [9] F. V. E. Hensling, H. Du, N. Raab, C.-L. Jia, J. Mayer, and R. Dittmann, "Engineering antiphase boundaries in epitaxial SrTiO₃ to achieve forming free memristive devices," *APL Mater.*, vol. 7, no. 10, p. 101127, Oct. 2019, doi: 10.1063/1.5125211.
- [10] T. Heisig *et al.*, "Antiphase Boundaries Constitute Fast Cation Diffusion Paths in SrTiO₃ Memristive Devices," *Adv. Funct. Mater.*, vol. 30, no. 48, p. 2004118, Nov. 2020, doi: 10.1002/adfm.202004118.

Reviewer #3 (Remarks to the Author):

The authors present an interesting study demonstrating a new approach to producing lamella devices for operando TEM study. Rather than GIS deposition to address the electrodes of the lamella, FIB cutting is used to introduce isolation of the electrodes prior to MEMS mounting. This produces a significantly reduced leakage current between the electrodes, with isolation of the electrodes also demonstrated via EBIC. As a case study, this preparation method allows the authors to observe atomic scale structural changes in their device during resistance switching. The authors also suggest that this paves the way for preparation of devices which more accurately reproduce macroscale behaviour.

I believe that the work shown in this manuscript is novel and valuable to the community, pending answers to a number of comments/queries I have detailed below. In general, I think it is reasonably well-written and that the conclusions are supported by the data. In addition to some minor comments, there are several discussion points which I do not feel are fully supported by evidence, and others which require clarification on exactly how measurements/simulations were carried out. So, I do not feel that any additional experimental work is necessary, but some well-considered changes to the manuscript certainly are in order to make it suitable for publication.

One concern is the quantity of supplementary material which is presented. There seems to be more supplementary content than main manuscript. Is it necessary for so much supplementary information to be present, or could the authors use more citations, for example?

We thank the reviewer for this suggestion. Now, in the revised manuscript, the number of supplementary figures was reduced from 29 to 23.

P15 L17 – high vacuum or actually UHV used?

We thank the reviewer to point out this mistake, within the TEM it is indeed UHV.

Fig. 1D – is this data for Pt and C the same as in Sup. Fig. 3? Was this device fabricated in the same way as a1 (aside from GIS)? The fabrication details for all samples relevant to the work should really be included. It is also important for the reader to understand exactly how the poor-performance/shorted devices were formed, to be clear that the process is standard GIS (i.e., there isn't an extreme excess of Pt or C deposited to strongly encourage shorted devices).

We agree with the reviewer that the information of how the process using standard GIS should be shown. However, to shorten the extent of the paper we have decided to refer our paper from 2017 by Zintler et al.,[1] where all the details of how the sample is prepared using GIS Pt-deposition is described.

Fig. 2 – the “switching” here at 18 V seems like an extremely high field, what voltage/polarity is required to switch back to the LRS? I would also disagree with the final sentence in the caption “Note the similar current density values of the TEM lamella device corresponding to the macro device during operation inside a TEM” – the current density in the lamella device at +/- 5 V is around 3×10^5 A/m². In the macro device the current density at +/- 5 V is around 10^4 A/m², which is significantly lower (more than an order of magnitude). In fact, the macro device is closer in current density to the HRS of the lamella.

In our experience, each TEM lamella of a memristor device is considered as a different device between each other, thus, they behave differently in terms of voltage required for switching. We believe this is either due to the different defect density encountered in each lamella or due to the

different carrier levels. Both scenarios are currently being studied in our group. Reason why we attribute operability of MIM devices based on the intrinsic electrical property of switching for memristors and the closely current densities obtained before switching or with other MIM devices such as tunable capacitors ((b1) and (b2) devices). Also, it is worth noting that to achieve such operando conditions together with visualization at high spatial resolutions in a TEM lamella is very challenging. In this regard, we were not able to cycle the sample shown in Fig 2. However, as a reference, cycling of another TEM lamella of the same STO-based memristor device (a1) was possible, and is shown in Supplementary Fig. 14A.

P3 L42 – assuming the authors are referring to the LRS of the lamella, the current at 5 V is more than 10 nA (closer to 100 nA) and the current density is more than 10^4 A/m² (closer to 3×10^5 A/m²), so do not agree that it is then reasonable to say on P4 L1 that lamella devices “despite their small dimensions, fully reproduce that of the corresponding macroscopic devices”. Perhaps the authors could reword this section to more accurately describe what the data are showing and indicating about their devices.

We agree with the reviewer about the terms to express proximity to the macro device’ current density may cause misunderstandings. We aimed to highlight the closely value achieved of the TEM lamella respect the macro device; the huge difference in current (more than 5 orders of magnitude) obtained with the GIS-based method and our approach led us to express the proximity in current densities as “similar”. However, we understand the reviewer point of view and we have changed the terms now to “closely” in order to avoid misunderstandings.

See page 4 line 4: “..., closely resemble.”

See page 13 Line 12: “..., closely achieved.”

Also sup. Fig. 8 caption makes a similar point as above on similarity between lamella and macro devices, but the plots show quite different behavior (different curve shapes). There is also not a clear set/reset occurring, there is only one distinct jump between states, under positive bias.

We thank the reviewer to point this out. The purpose figure was to compare the data from Cho et al., [2] with our TEM lamella device. We agreed with the reviewer that the switching in the macro device was not fully obtained by Cho et al., but in order to resemble what was obtained in the macro device we have also, in purpose, not achieved a fully switching behavior in our TEM lamella device that is shown in sup. Fig. 8B. However, the TEM lamella of such device can indeed be switch and cycle properly in a TEM lamella device, this has been demonstrated in sup. Fig. 14B.

The resistance switching of the lamella device e.g. sup fig 14 actually looks better than the macro device e.g., sup fig 13, which looks like it is showing a probe contact/charging artefact (HRS min current not at 0 V). Additionally, the switching to LRS is at a high voltage, so possibly this is just hard breakdown (no reset shown)? Can the authors comment on why their lamella device shows better switching behavior than the macro devices? No info is given on how the macro devices were operated electrically so it’s possible there was an issue with the instrumentation. Was the switching of the macro devices done between the S and G locations as depicted in Sup. Fig. 20?

We greatly appreciate the expertise of the reviewer in this field, indeed as the reviewer mentions and as described in the materials and method section, only the (b1) and (b2) devices were tested as described in sup. Fig. 20. The (a1) device was grown without patterning so the electrical characterization described in sup. Fig. 20. was not possible. Therefore, we have used a nano probe station to test the device electrically as shown in sup. Fig. 22. Indeed, as the reviewer point out, the contacts with the nano probe station were not ideal and might cause probe contact artifacts.

However, the goal of device' switching was achieved. In the TEM lamella the contact is more stable, therefore, the switching process showed a better behavior.

Why are a2, b1 and b2 included in the table/fabrication details? They are not mentioned anywhere in the manuscript, only in the supplementary information. Is it necessary to include them in the table or supplementary information if they are not relevant to the main study and its conclusions?

We believe that Table 1 holds significance as it showcases the wide-ranging applicability of the sample preparation routine described in this study across various MIM devices. This helps illustrate that this routine isn't limited solely to memristor devices but is also applicable to other types of MIM systems and materials. However, in consideration of readability, we concur with the reviewer's suggestion and have relocated the table to the supplementary material.

Following the above comment, it would be good to gauge the repeatability of the process. Is all the data from a single device/fabrication (i.e., sample a1) or the result of multiple fabrications? How widely applicable a solution is the authors' method to addressing the shortfalls of standard GIS lamella preparation? Is it significantly more difficult to get the FIB cuts correct vs doing GIS deposition? (Of course, I appreciate the huge challenge in preparing even a single lamella device!).

The purpose of Figure 1H, is to show the repeatability of the TEM lamella process obtained with one of the devices (a1). Additionally, but not included in this manuscript, our experience during the last years using this method has showed repeatability in all the other devices as well. The current densities are reproducible, but the forming voltage of memristors may vary between lamellae.

About how difficult our approach is compared with the GIS deposition method. In general words, the workflow here has been simplified, and thus preparation is faster, nevertheless user experience dependent. The workflow shown below was obtained based on the experience of a particular FIB user, in this specific case by the first author of this article, O. Recalde-Benitez. It must be considered that the sample preparation times would drastically change depending on the FIB equipment, the used micromanipulator, and the materials (including electrodes) involved in the whole device heterostructure or two-terminal device. Therefore, this table must be considered only as a reference and not as a rule of thumb for every user. Here, both approaches have been tested using the same FIB (JEOL JIB-4600F) and MEMS chips (DENSolutions Lightning chip). It is deemed by the authors that our actual FIB sample preparation routine reduces the routine time of our previous method for at least half of the time. This is mainly due to the absence of FIB deposit paths for electrical contacts and the simpler in-micromanipulator thinning process.

The success rate of TEM lamellae attachment on the MEMS chips with the current approach is relative to the user experience. In general, the GIS-based approach is complex due to the small angle gap that exists between the TEM lamella and the chip when the GIS system is introduced in our JEOL-FIB geometry (approx. 15 degrees). The deposition of Pt (as the first attachment point) is usually weak; therefore, the sample tends to fly out when the GIS system is removed. The chip could also break apart due to the short distances between the micromanipulator, the GIS system, and MEMS chip geometry. On the other hand, our current approach, which is based solely on the use of van der Waals forces and thus, does not require any GIS process. Therefore, the complete TEM lamellae attachment process yields a much higher success rate.

GIS approach

Current approach

P4 L10 – “In short-circuited samples, the electric field is distributed discontinuously across the lamella resulting in non-operative devices (Supplementary Fig. 5)” Where is the evidence supporting this? Sup. Fig. 5 just shows a device with some damage after the application of a relatively high current. In what way is this indicative of a discontinuous field distribution, and has this distribution been measured or simulated in any way? The figure also shows a maximum field of 20 V/um, which for an a1 device (105 nm thick active layer) implies a maximum voltage of 190 V (which is far beyond any normal operational voltage for a resistance switching device), not 2 V, as is shown on the top x axis. Which scale is correct?

The presence of leakage current pathways formed by Pt or C contamination directs the electrical current along these contaminated regions. Regardless of the applied voltage, if the current does not flow through the dielectric layer of the TEM lamella device, resistive switching in memristors cannot occur. For example, as shown in Supplementary Figure 5, a memristor TEM lamella device in a short-circuit state experiences very high currents (0.02 A) at relatively low voltages (2 V) due to Pt contamination, which leads to localized melting points on the TEM lamella. However, switching did not occur before layer damaged. Furthermore, increasing the voltage will raise the current levels without leading into a change in resistive state but potentially resulting in the failure of the sample and the MEMS chip itself.

A note has been added in the manuscript to highlight this important suggestion from the reviewer:

The change to the manuscript text is shown below:

Page 2 lines 46-47: “...., affecting not only the electric responses of MIM devices but also”

Furthermore, SEM EBIC in a TEM lamella contacted with Pt paths is displayed below. At the top is a SEM image, and at the bottom is the corresponding EBIC image. SEM EBIC is a widely recognized technique for mapping electric fields. Therefore, the red contrast in this image highlights the non-uniform electric field to which a lamella is exposed.

How can the authors be sure that there is no drift between the images in Fig. 3E and F?

Sample drifting or tilting could indeed happen during an in situ TEM biasing experiment. However, it is not the case of the experiment shown in figure 3. To cross-validate this, HAADF-STEM imaging simulations were carried out using an ideal SrTiO₃ supercell (without extended defects) with sample tilting of 0.05, 0.1, 0.3, 0.5, 1 and 5 degrees from [110] towards [1-10] and [001], respectively. In all the simulated HAADF-STEM images (shown below), there is no observable extra atomic columns. Therefore, we believe that the observed extended defects in Fig 3F are not due to sample tilting. For the details of the image simulation algorithms implemented, please refer to the supplementary section.

P5 L35 – “A detailed and systematic comparison between current and current densities of millimeter-sized thin-film-based devices and their counterpart TEM lamellae is a viable and indispensable way to assure an appropriate electrical contact for further in situ/operando TEM experiments.”

P18 L25 – how was the CASINO simulation carried out? There needs to be some more detail in order to support the statement that “Absorbed currents, depending on material density, could be considered negligible.”

For the CASINO simulation, an electron beam probe of 0.08nm in diameter with a spacing of 0.1nm, 1000 electrons at 200 keV were simulated considering a line scan over a stack material system

consisting of C/Pt/STO/STO:Nb with a thickness of 100nm. Additionally, secondary electrons were generated. Moreover, the co-author of this manuscript, Dr. W. A. Hubbard has demonstrated that absorbed current can be considered negligible in STEM SEEBIC in his paper from 2018. Some other references have also proven that the contribution of absorbed electron is negligible for electron transparent (S)TEM samples.

Nevertheless, due to the extent supplementary information in this manuscript and following the suggestions from the reviewers we have decided to remove some supplementary. The CASINO simulation is not necessary for this study; therefore we have removed it.

Sup. Fig. 28 – what is the purpose of this Figure? It doesn't seem to be relevant to the manuscript and just shows potential chips for other applications, without any discussion. Is the purpose here to indicate that various GIS-free chip arrangements are possible, in which case are further experiments/figures/discussion required to demonstrate the flexibility of the lamella fabrication process?

The idea of this figure was to highlight the chips designs that has been used for this manuscript. However, we agree with the reviewer that the image might not be contributing with the manuscript, therefore, we have decided to remove it.

References

- [1] A. Zintler *et al.*, "FIB based fabrication of an operative Pt/HfO₂/TiN device for resistive switching inside a transmission electron microscope," *Ultramicroscopy*, vol. 181, pp. 144–149, Oct. 2017, doi: 10.1016/j.ultramic.2017.04.008.
- [2] S. Cho *et al.*, "Self-assembled oxide films with tailored nanoscale ionic and electronic channels for controlled resistive switching," *Nat. Commun.*, vol. 7, no. 1, Nov. 2016, doi: 10.1038/ncomms12373.
- [3] L. Molina-Luna *et al.*, "Enabling nanoscale flexoelectricity at extreme temperature by tuning cation diffusion," *Nat. Commun.*, vol. 9, no. 1, p. 4445, Oct. 2018, doi: 10.1038/s41467-018-06959-8.

Reviewers' comments:

Reviewer #1 (Remarks to the Author):

The authors clarified the technical concerns.

Reviewer #2 (Remarks to the Author):

Thanks you very much for the answers and the corresponding modifications of the manuscript. The paper can definitively be recommended to be published in Communications Engineering.

Reviewer #3 (Remarks to the Author):

I'd like to thank the authors for their detailed responses to my review, and for the changes that they have made to the manuscript. I only have a few remaining points, and they mainly concern some aspects of the language used; the accuracy of some of the discussion/description of figures/data should be improved to make sure that the message is clear.

I appreciate the authors' changes to include "closely resemble" (p4 l2) and "closely achieved" (p13 l11), although I would argue that these phrases are still not appropriate to convey the author's meaning, and the figures do not show what I would consider to be "similar", a "close resemblance" or a "close achievement". I am not criticising the quality of the data or the technique, but rather the accuracy of describing what is being shown. As I understand it from the rebuttal, the authors want to highlight that the lamella devices are reasonably comparable to the macroscale devices, despite the inherent variation in the electrical properties resulting from lamella device fabrication. So, I would recommend saying something like that, or otherwise removing these sort of statements as they seem to be confusing and not really necessary to the discussion.

My comment on Sup. Fig. 8 has not been addressed (I couldn't see any revision to the supplementary information). I am not criticising the data or the technique, but I think the authors need to be more accurate than just using the word "similar". To me, the switching does not appear similar. The orders of magnitude are different, the shape of the curves is different, the voltage ranges are different. Please could the authors either expand on "similar" e.g., by either pointing out the similarities between A and B, or remove the statement on similarity (is it actually necessary/helpful to mention the proposed similarity?).

I'm afraid I don't follow the authors' response to my comment on how the macro devices were tested electrically. The authors mention that the a1 sample was not patterned, but it is a Pt/STO/Nb:STO device so it has a patterned top electrode. They also mention that characterisation as in sup Fig. 20 was not possible, but the caption of sup Fig. 20 mentions that this technique was used for a1 devices.

I appreciate the authors' detailed response on repeatability and process difficulty, and apologise that I

missed the message in Fig. 1H on my first review. If they agree, I would encourage the authors to briefly include some of this additional information on observed workflow to strengthen the message of the manuscript, that their technique is a significant improvement (e.g., it is an important finding that the prep time is halved, and that eliminating GIS improves the yield due to a reduction in mechanical challenges).

I thank the authors for their response on my comment P4 L10 – “In short-circuited samples, the electric field is distributed discontinuously across the lamella resulting in non-operative devices (Supplementary Fig. 5)”. I think here the issue is the word “discontinuously”. In their response, the authors in fact use the phrase “non-uniform”, which I think would be much more appropriate to use, e.g., something like “In short-circuited samples, the electric field might be distributed non-uniformly across the lamella, resulting in damaged, non-operative devices.”

On the shifting during biasing, if the image shifted by a full column, would it be possible to know that this had occurred?

Reviewer #3 (Remarks to the Author):

I'd like to thank the authors for their detailed responses to my review, and for the changes that they have made to the manuscript. I only have a few remaining points, and they mainly concern some aspects of the language used; the accuracy of some of the discussion/description of figures/data should be improved to make sure that the message is clear.

I appreciate the authors' changes to include "closely resemble" (p4 |2) and "closely achieved" (p13 |11), although I would argue that these phrases are still not appropriate to convey the author's meaning, and the figures do not show what I would consider to be "similar", a "close resemblance" or a "close achievement". I am not criticising the quality of the data or the technique, but rather the accuracy of describing what is being shown. As I understand it from the rebuttal, the authors want to highlight that the lamella devices are reasonably comparable to the macroscale devices, despite the inherent variation in the electrical properties resulting from lamella device fabrication. So, I would recommend saying something like that, or otherwise removing these sort of statements as they seem to be confusing and not really necessary to the discussion.

We greatly thank the reviewer, and agree to follow his helpful suggestions, by adding a description in (P4|6) and removing the sentence regarding similarity in (p12|11).

P4/6: "These results indicate that the electrical performance of the fabricated TEM lamella nanodevices, though their small dimensions, closely resemble that of the corresponding macroscopic devices, despite the inherent variation in the electrical properties due to the FIB device lamella fabrication."

My comment on Sup. Fig. 8 has not been addressed (I couldn't see any revision to the supplementary information). I am not criticising the data or the technique, but I think the authors need to be more accurate than just using the word "similar". To me, the switching does not appear similar. The orders of magnitude are different, the shape of the curves is different, the voltage ranges are different. Please could the authors either expand on "similar" e.g., by either pointing out the similarities between A and B, or remove the statement on similarity (is it actually necessary/helpful to mention the proposed similarity?).

We agree with the reviewer, the statement of similarity has been completely removed.

I'm afraid I don't follow the authors' response to my comment on how the macro devices were tested electrically. The authors mention that the a1 sample was not patterned, but it is a Pt/STO/Nb:STO device so it has a patterned top electrode. They also mention that characterisation as in sup Fig. 20 was not possible, but the caption of sup Fig. 20 mentions that this technique was used for a1 devices.

We thank the reviewer for point this out, and we apologize if our previous response was unclear. The pattern we referred to pertains to the outer and inner circular patterns described in Fig. 20. These patterns are designed to facilitate proper electrical contacting

using dedicated probe stations for measuring the electrical response of the macro-sized devices¹. However, the (a1) device was not patterned in this manner, and as a result, we were unable to test its electrical response in the same way as the (b1) and (b2) devices. We appreciate the reviewer once again for bringing this to our attention. We have now corrected Fig. 20 to only include the (b1) and (b2) devices and P16|L34.

I appreciate the authors' detailed response on repeatability and process difficulty, and apologise that I missed the message in Fig. 1H on my first review. If they agree, I would encourage the authors to briefly include some of this additional information on observed workflow to strengthen the message of the manuscript, that their technique is a significant improvement (e.g., it is an important finding that the prep time is halved, and that eliminating GIS improves the yield due to a reduction in mechanical challenges).

We greatly appreciate the reviewer for these very helpful suggestions to further improve the manuscript. We have now included a sentence in P3|L26-28 that highlights the time reduction in sample preparation.

P3|L26-28: "The time required to prepare a sample was significantly reduced, approximately half of the time, thanks to the elimination of steps related to sample attachment and the establishment of a conductive path using the GIS system."

I thank the authors for their response on my comment P4 L10 – "In short-circuited samples, the electric field is distributed discontinuously across the lamella resulting in non-operative devices (Supplementary Fig. 5)". I think here the issue is the word "discontinuously". In their response, the authors in fact use the phrase "non-uniform", which I think would be much more appropriate to use, e.g., something like "In short-circuited samples, the electric field might be distributed non-uniformly across the lamella, resulting in damaged, non-operative devices."

Thanks for this comment, we agree with the reviewer and now the term has been modified in the main manuscript.

On the shifting during biasing, if the image shifted by a full column, would it be possible to know that this had occurred?

We appreciate the reviewer's question. Shifting by one unit cell results in a lattice distortion. In the figure below, we illustrate both a partial atom column shift and a full column shift. Figure A depicts the lattice without any shifting of atom columns. When observing this lattice in the direction indicated by the blue arrow [100], we observe a periodic arrangement of the atoms. In Figure B, we show a partial shifting of an atom column (orange atoms) in the [010] direction. When observed in the [100] direction, this shifting appears as a second atom column next to the first one. Finally, in Figure C, we illustrate a full atom column shift in the [010] direction. In this case, a lattice distortion may occur due to the shifting of the entire

column.

Reference

1. Walk, D. *et al.* Characterization and Modeling of Epitaxially Grown BST on a Conducting Oxide Electrode. in *2018 48th European Microwave Conference (EuMC)* 563–566 (IEEE, 2018). doi:10.23919/EuMC.2018.8541771.